# 3D Structural Insights into β-Glucans and Their Binding Proteins

**DOI:** 10.3390/ijms22041578

**Published:** 2021-02-04

**Authors:** Noriyoshi Manabe, Yoshiki Yamaguchi

**Affiliations:** Division of Pharmaceutical Physical Chemistry, Tohoku Medical and Pharmaceutical University, 4-4-1 Komatsushima, Aoba-ku, Sendai, Miyagi 981-8558, Japan; manabe@tohoku-mpu.ac.jp

**Keywords:** β-glucan, 3D structure, triple helix, X-ray crystallography, NMR, βGRP/GNBP3, Dectin-1, endoglucanase

## Abstract

β(1,3)-glucans are a component of fungal and plant cell walls. The β-glucan of pathogens is recognized as a non-self-component in the host defense system. Long β-glucan chains are capable of forming a triple helix structure, and the tertiary structure may profoundly affect the interaction with β-glucan-binding proteins. Although the atomic details of β-glucan binding and signaling of cognate receptors remain mostly unclear, X-ray crystallography and NMR analyses have revealed some aspects of β-glucan structure and interaction. Here, we will review three-dimensional (3D) structural characteristics of β-glucans and the modes of interaction with β-glucan-binding proteins.

## 1. Introduction—Variation of β-Glucan Primary Structures

β-glucan is widely distributed in bacteria, fungi, algae and plants. β-glucan can be utilized as a “recognition pattern” in host defense systems. To understand the system recognizing β-glucan, the primary and tertiary structures of β-glucan need to be elucidated as well as the cognate receptors identified. β(1,3)-glucan is a polymer with the main chain composed of β(1,3)-linked D-glucose and often branched with side chains, starting with a β(1,6)-linked D-glucose residue. The length of the main chain, the interval between branching and the side chain length (structure) are characteristic of each β-glucan from different sources (Figure 1). Curdran, which was isolated from an *Agrobacterium* species, is a linear β(1,3)-glucan essentially free from β(1,6) branching, with its degree of polymerization (DP) approximately 135 glucose units [1,2,3]. Curdran is water-insoluble and tends to form a gel upon heating. By contrast, schizophyllan, which is produced by *Schizophyllum commune*, is a β(1,3)-glucan modified with β(1,6)-linked monoglucose residues every three β(1,3)-linked glucose residues and has a molecular weight of 4.3 × 10^6^ Da [4]. Unlike curdran, schizophyllan is water-soluble, suggesting a role for β(1,6)-linked monoglucose residues in defining the solubility of β-glucan. Scleroglucan, produced by *Sclerotium* species, is very similar to schizophyllan in having β(1,6)-linked monoglucose residues at about every three β(1,3)-linked glucose residues [5]. Lentinan, from *Lentinus edodes*, is a β-glucan having two β(1,6) glucopyranoside branches for every five β(1,3)-glucose residues [6]. A few internal β(1,6) linkages may be present as β(1,6)-linked side chains. Overall, the chemical structure of lentinan is similar to that of schizophyllan and scleroglucan. Laminarin, which was isolated from *Laminaria digitata*, is a water-soluble, small β-glucan with DP of 20~30. Like schizophyllan, laminarin has an average 1.3 β(1,6)-linked monoglucose residues per molecule (one branched glucose residue is covalently attached every seven β(1,3)-linked glucose residues). The reducing end of laminarin is capped with reducing D-glucose (G-series) or non-reducing D-mannitol (M-series) in a ratio of 1:3 [7]. Yeast cell wall β-glucans show different branching structures. A soluble *Candida* species β(1,3)-glucan (CSBG), a dimethyl sulfoxide-soluble fraction extracted from the NaClO-oxidized cell wall, is modified with long β(1,6)-linked glucosyl side chains [8]. The ratio of β(1,3):β(1,6) glucosyl linkages varies according to the *Candida* species, ranging from 1:0.2 to 1:0.7, and the DP of the side chains may be as few as 10 to over 50. Many other β-glucans are known with differences in backbone chain length, branching interval and side-chain structures. It should be emphasized that differences in chemical structure of the β-glucans likely define tertiary structure and hence affect biological activities [9].

## 2. 3D Structure of β-Glucan—What Does β-Glucan Look Like?

Accumulating evidence suggests that long-chain β-glucan forms a triple helix structure similar to that of collagen. X-ray diffraction studies of several β(1,3)-glucans reveal a triple helical backbone structure. Probable models of lentinan have been proposed from X-ray fiber diffraction and theoretical conformational analysis [10]. Five models have been proposed, one is a single helix, two are double helices and two are triple helices (right-handed or left-handed). Of these, the most probable is the right-handed triple helix, by analogy with the determined structure of β(1,3)-xylan. Hydrated curdran is a triplex of right-handed, six-fold helical chains [11]. The individual chains of curdran are composed of a six-glucose unit per turn. Water molecules are clustered near the O4 and O5 oxygen atoms of the glucose residues in curdran, and this may indicate the presence of water-mediated hydrogen bonds between the two glucose residues. X-ray diffraction experiments on curdran and scleroglucan show the diameter of the triplex to be 14.3 Å and 17.3 Å, respectively [12]. Each chain is composed of six glucose residues per turn, yielding a pitch of 2.9–3.0 Å per residue. These parameters indicate that the backbone conformation of scleroglucan is similar to that of curdran. It is therefore likely that the β(1,6)-linked side chain of scleroglucan is outside of the triple helix, such that it does not significantly disturb the backbone conformation.

From these diffraction data, atomic models of triplex β-glucan have been proposed. In a widely accepted model, the hydroxy group at position 2 (2-OH) of each glucose residue forms interchain hydrogen bonds with the 2-OH groups in the other two strands (Figure 2). These are formed perpendicular to the axis of the triple helix. The 2-OH groups of the glucose residues lie inside the hydrophobic core. In contrast, the hydroxy groups at position 6 (6-OH) of the glucose residues face towards the hydrophilic solvent. Thus, it follows that β-glucan branching occurs at the 6-position of a main-chain glucose residue.

In addition to X-ray fiber diffraction analysis, several methods have been applied to probe the structure of triple helical β-glucans. Solid-state ^13^C-NMR spectroscopy has been applied to conformational analysis of β-glucans. It has been shown that ^13^C chemical shifts readily distinguish three conformations, single chain, single helix and triple helix. The triple helix has also been characterized in solution state. Solution studies of shizophyllan by light scattering and viscosity measurement suggest a semi-flexible, rod-like structure with the pitch per glucose residue and the persistence length of the triple helix of 3.0 A and 18 Å, respectively [13].

What is the required length of β-glucan for formation of a triple helix? To answer this question, β-glucans with different chain length were obtained from partial hydrolysis of curdran and examined by optical rotatory dispersion [14]. A β-glucan chain with DP more than 200 (molecular mass of 32,000 Da per chain) was found to be necessary to form an ordered structure. β-glucans with DPs below 25 are soluble and assume a disordered structure in water. For schizophyllan, a molecular mass of higher than 50,000 Da (as a triplex) is required for the formation of a triple helix [15]. Laminarin from *Laminaria digitata*, whose molecular mass is around 5000 Da, is present mostly in a monomeric form with triplex structures a minor population (5%) [16,17]. According to these studies, it can be seen that a molecular weight of more than several tens of thousands per one β-glucan chain is necessary to form a stable triple helix. 

Triple helices of β-glucan denature into random coils when dissolved in an alkaline solution (pH > 12) [12], in dimethyl sulfoxide (DMSO) [18] or when the temperature is increased above the melting temperature (~135 °C) [19]. The mechanisms are different in each case. It is likely that in an alkaline condition, hydroxyl groups will be negatively charged, which would lead to electrostatic charge repulsion between the strands [9]. DMSO will destabilize hydrogen bonds and high temperatures are expected to destabilize strands [9]. Renaturation of β-glucans from the denatured state has also been examined. Denatured schizophyllan dissolved in DMSO and dialyzed against water renatures to a mixture of circular, linear and aggregated structures [20]. Denatured lentinan can be renatured by dialysis against water after denaturation in 0.15 M NaOH [21]. From AFM observations, renatured lentinan consists of linear, circular and branched species of triple helix.

## 3. Dectin-1–β Glucan Interaction

Dectin-1 is the most studied of the β-glucan receptors from vertebrates [22]. It is mainly expressed on myeloid cells such as macrophages, dendritic cells and neutrophils. Dectin-1 is a type II membrane protein composed of an extracellular lectin domain, a transmembrane domain and a cytosolic region with an immunoreceptor tyrosine-based activation motif called ITAM. The lectin domain is responsible for Ca^2+^-independent β-glucan binding, and binding depends on chain length. The minimum length required for detectable binding is a 10- or 11-mer, as determined by glycan microarray experiments [23]. In these experiments, there was no binding of glycans other than β-glucan, indicating high specificity. An NMR interaction study determined that Dectin-1 binds weakly with laminarihexaose (degree of polymerization, DP = 6), moderately with a chemically synthesized β-glucan chain (DP = 16) and strongly with laminarin (average DP = 25) [16]. STD-NMR analysis using β-glucan chains (DP = 6, 16 and 25) revealed that only the middle part of the β-glucan chain is recognized, not the reducing/non-reducing ends [16,24]. The mechanism of the chain-length-dependent interaction is unclear. NMR data suggest that increasing β-glucan chain length correlates with increasing secondary structure formation. In general, a polymer such as a polypeptide can form secondary structures anywhere along its length, except at the termini where there is no hydrogen-bonding partner. Certainly, laminarin, with an average DP of 25, does have secondary structure, as evidenced by deuterium-induced ^13^C-NMR isotope shifts [16]. The evidence then points to chain-length-dependent interaction being explained by the presence of helical structures, which are expected to snugly fit into the ligand-binding site of the Dectin-1 lectin domain. 

In addition to the backbone chain-length of β-glucan, β(1,6)-branching affects binding to Dectin-1. Adams et al. investigated the interaction of Dectin-1 with a library of natural and synthetic β-glucans [25]. Dectin-1 differentially interacted with β-glucans over a wide range of affinities. The range of IC_50_ is from 2.6 mM for nonbranched, linear octasaccharide glucan, to an astounding 2.2 pM for glucan phosphate. This is likely the highest affinity interaction reported for a C-type lectin-like receptor. Importantly, the branched nonasaccharide (ID_50_ = 2.9 μM) is 1000-fold stronger than linear nonasaccharide (ID_50_ = 2.6 mM). There was a ~270-fold increase in affinity between the branched nonasaccharide and linear decasaccharide (0.7 mM versus 2.9 μM). It remains to be seen how the Dectin-1 lectin domain preferentially interacts with branched β-glucan.

A crystal structure of the murine Dectin-1 lectin domain has been reported in the ligand-free form (Figure 3) [26]. The lectin domain shows a typical C-type lectin fold composed of two anti-parallel β-sheets and two α-helices, with two coordinated Ca^2+^ ions. The bound Ca^2+^ ions are not required for ligand-binding but stabilize the structure of the domain. There is a dimeric arrangement in the crystal lattice, which trapped a short β-glucan ligand at the interface. The site is far from the putative ligand binding site previously defined by Trp221 and His223 [27]. The dimer found in the crystal will not be formed under physiological conditions because Asn185 is normally N-glycosylated [28] and located in the dimer interface. Currently, a three-dimensional (3D) structure of a β-glucan–Dectin-1 complex is not available, and therefore neither is the β-glucan structure nor binding mode at the atomic level.

The lectin domain of Dectin-1 forms higher-order oligomers when bound to laminarin [26]. Oligomer formation is cooperative, with a Hill coefficient of ~3 [29]. How the β-glucan directs the cooperative oligomer formation is not known and awaits future study. The ligand-induced oligomer formation of Dectin-1 may occur at the cell surface in the physiological situation, and the oligomerization may enhance the signaling through the cytosolic region. A recent preprint shows that full-length Dectin-1 on the cell surface forms a dimer/oligomer upon binding to structured β-glucans [30].

Dectin-1 has always been considered a β-glucan receptor participating in the innate immune self-defense system, but recent reports suggest other functional aspects. Dectin-1 apparently binds to the conserved core domain of annexins (annexin A1, A4 and A13) expressed on apoptotic cells and induces immune tolerance [31]. The binding is very strong at nanomolar affinity via a site distinct from the β-glucan interaction site. As expected, Dectin-1-deficent mice generate a stronger immune response against apoptotic cells and develop autoimmunity. Another study suggests that Dectin-1 recognizes the N-terminal asparagine at the glycosylation site as well as the core fucose on the N-glycan of the IgG-Fc region [32]. Dectin-1 also appears to be involved in the recognition of characteristic N-glycans on antibodies, although its biological significance remains unclear. Further study will likely show that Dectin-1 has a whole range of functions in vivo.

## 4. Complement Receptor 3(CR3)–β-Glucan Interaction

Complement receptor 3 (CR3, Mac-1, α_m_β_2_ integrin, CD11b/CD18) is a heterodimeric complex composed of α_m_ integrin (CD11b, 165 kDa) and β_2_ integrin (CD18, 95 kDa), expressed on mature myeloid cells, NK cells and minor subsets of B and T cells. CR3 binds to complement component iC3b and is responsible for phagocytosis of complement-opsonized particles. Both chains of CR3 are multi-domain proteins and several studies have reported on the lectin domain and sugar-binding specificity. Several lines of evidence suggest that a carbohydrate binding site is likely located in the C-terminus of CD11b [33,34]. CR3 binds a variety of carbohydrate ligands, including β-glucans but not α-mannan [33]. The 3D structure of the lectin domain and the binding mode is currently not known, precluding the understanding of its physiological function.

## 5. βGRP/GNBP3–β-Glucan Interaction

β1,3-Glucan recognition protein (βGRP)/Gram-negative bacteria-binding protein 3(GNBP3) is a soluble pattern recognition receptor found in the hemolymph of invertebrates such as silkworm and Drosophila. βGRP/GNBP3 is one of the best characterized families of pattern recognition receptors in invertebrates [35]. βGRP/GNBP3 binds to long, structured β(1,3)-glucan [36,37]. βGRP/GNBP3 consists of two domains: a well-conserved N-terminal domain and a C-terminal glucanase-like domain which does not have glucanase activity. The N-terminal domain consists of about 100 amino acid residues and binds β(1,3)-glucan. Binding of βGRP/GNBP3 to β(1,3)-glucan through its N-terminal domain triggers an innate immune response by activation of the Toll pathway.

In 2009, pioneering structural work revealed the 3D structures of the ligand-free βGRP/GNBP3 N-terminal domain by X-ray crystallography [37] and solution NMR spectroscopy [36]. The crystal structure of the Drosophila βGRP/GNBP3 N-terminal domain shows an immunoglobulin fold. The β-glucan binding site was predicted to be the hydrophobic surface which is masked by a C-C’ loop. A ligand-binding mechanism was proposed, in which long-chain structured β-glucan binds to the hydrophobic surface with displacement of the occluding C-C’ loop. At about the same time, the solution structure of the silkworm βGRP/GNBP3 N-terminal domain also showed the same β sandwich fold. The NMR titration experiments and mutational analysis suggested that β-glucan preferentially binds to the non-aromatic concave surface. Further studies then described crystal structures of the N-terminal β(1,3)-glucan recognition domain of βGRP/GNBP3 from *Plodia interpunctella* and *Bombyx mori* in complex with β(1,3)-linked glucose hexamer (laminarihexaose) (Figure 4). In both complexes, the laminarihexaoses are spatially arranged to form pseudo-quadruplex structures, which well mimics the triplex of β-glucan. These 3D structures can be utilized for understanding the β-glucan triplex and its interaction with protein receptors. The laminarihexaoses form inter-strand hydrogen bonds between 2-OH groups. The observed hydrogen-bonding pattern is very similar to that in models of triple helical β-glucan. In addition, the helical structure of the laminarihexaoses is stabilized by intra-strand and inter-residue hydrogen bonds between their OH-4 (i-1) and O5(i), which is also found in the triplex model. Furthermore, water-mediated hydrogen bonds are common between OH-4 (i-1) and O6(i). The binding site of laminarihexaoses is on the convex surface, which is different from previously proposed sites [36,37]. The reason for the discrepancy is not known, but one possibility may be differences in chemical structures of the β-glucans used in the various experiments. Another possible reason may stem from site-directed mutagenesis affecting the local structure of the protein.

The interaction mode of laminarihexaoses with βGRP/GNBP3 is characteristic: one βGRP/GNBP3 interacts with three laminarihexaoses, which are spatially arranged like a triple helix. This interaction mode well explains the previous observations that βGRP/GNBP3 binds triple-helical β-glucan strongly but has little affinity for denatured β-glucan or shorter β(1,3)-linked glucan chains [36,37]. This is the first example that provides a structural basis for how triple helical β(1,3)-glucan is recognized by a protein receptor at the atomic level.

βGRP/GNBP3 is widely distributed among species and forms a large family [39]. A recent study reports on the ligand binding characteristics of βGRP/GNBP3 from four insects (*Bombyx mori*, *Plodia interpunctera*, *Tribolium castaneum* and *Tenebrio molita*) [40]. From the solid-phase ELISA assays, the binding specificities were found to be categorized into two groups. One group (*Bombyx mori* and *Plodia interpunctera*) tend to bind to triple-helical native β-glucans, while the other group (*Tribolium castaneum* and *Tenebrio molita*) prefer alkaline-treated β-glucans. These results suggest that the preferred β-glucan conformation is different for individual βGRP/GNBP3. The difference will be clarified through 3D structural analysis of each βGRP/GNBP3.

βGRP/GNBP3 binding to triplex β-glucan activates the prophenoloxidase cascade [41,42] and antifungal Toll pathway [43]. The mechanism is not fully understood. Interaction of β-glucan laminarin (~6 kDa) with βGRP/GNBP3 N-terminal domain (15 kDa) has been analyzed by solution NMR and analytical ultracentrifugation [44]. The N-terminal domain of *Plodia interpunctella* βGRP GNBP3 is sufficient to activate the prophenoloxidase pathway, resulting in melanin formation. The N-terminal domain forms a stable complex with laminarin (~102 kDa) and is possibly composed of six proteins and three laminarins. The ligand-induced self-association of βGRP/GNBP3 may provide a platform for recruitment of downstream proteases. Similarly, the βGRP/GNBP3 N-terminal domain of *Manduca sexta* also provokes oligomer formation. When the laminarin/protein ratio is low (ca. 1), an insoluble aggregate forms, when high (>5), a soluble complex containing at least five protein molecules results. It appears from these reports that ligand-induced oligomer formation of βGRP/GNBP3 is the initial event that then triggers the downstream pathway. The triple helix structure of β-glucan likely plays a key role in the oligomer formation, but this needs further structural analysis.

## 6. Factor G–β-Glucan Interaction

Horseshoe crab Factor G is a non-covalent heterodimer composed of anα-subunit (72 kDa) and a β-subunit (37 kDa). The α-subunit is responsible for the recognition of β-glucan, while the β-subunit is a serine protease which becomes activated when factor G binds to β-glucan. The α-subunit comprises three kinds of modules: a single β-glucanase A1-like module, three tandem xylanase A-like modules and two tandem xylanase Z-like modules (Z1 and Z2). Z1 and Z2 modules have independent β-glucan-binding sites and cooperatively enhance avidity toward β-glucan-containing pathogens [45]. A chemical shift preservation experiment has helped map the laminaripentaose-binding site to a cleft on a β-sheet in the predicted 3D model [46]. Activation of Factor G increases over 100-fold on treatment of β-glucan with 0.3 M NaOH, which converts a triple helix to a single helix [47]. The binding specificity and activity of Factor G has led to its clinical application as a diagnostic reagent for the detection of fungal infections in humans [48].

## 7. Other β-Glucan–Protein Interactions

In addition to these β-glucan receptors, some other related proteins are known to bind structured β-glucan. In the CAZy database (Carbohydrate Active Enzymes database), carbohydrate-binding module (CBM) families 4, 6, 13, 32, 39, 43, 52, 54, 56, 65, 72, 76, 79, 80, 81 and 85 have an ability to interact with β-glucans [49,50]. Some 3D structural information is available on how CBM binds to β-glucan chains. A good example is the β(1,3)-glucanase BH0236 from *Bacillus halodurans*, which is a multidomain protein composed of three parts: a N-terminal family 81 glycosyl hydrolase (GH81) catalytic module [51], an internal CBM6 that binds to the non-reducing end of β(1,3)-glucan chains [52] and a C-terminal CBM56 that binds to β(1,3)-glucan chains [53] (Figure 5). There are crystal structures of a *Bacillus halodurans* endo β-glucanase (GH family 81) catalytic domain in complex with β-glucan chains [51]. Soaking crystals with laminarin resulted in three β-glucan chains binding to the protein. A large oligosaccharide was found in the active site that was modelled as laminaridecaose (10-mer). In addition to this, two other β-glucan chains (DP = 2 and 3) were detected close to the bound laminaridecaose. Interestingly, the structure of laminaridecaose and other shorter oligosaccharides roughly mimic the triple helical β-glucan. The architecture of the catalytic site in this enzyme seems structured to accommodate the double and/or triple helical quaternary structures of β-glucan chains. The crystal structure of C-terminal CBM56 is similar to the N-terminal domain of *Plodia interpunctella* GNBP3, with RMSD of 1.9 Å using Cα atoms. The NMR-mapped laminarin-binding site of this C-terminal CBM56 corresponds to the surface of the laminarihexaose-binding site in *Plodia interpunctella* GNBP3 [38]. They may share a common β(1,3)-glucan binding mode.

Another endo β-glucanase (GH64), from *Paenibacillus barengoltzii*, is also capable of recognizing triplex β-glucan. This β-glucanase is composed of two regions: a N-terminal CBM56 with β-glucan binding ability, and a C-terminal region corresponding to a β-glucanase domain (GH64). A crystal structure of this full-length enzyme (inactive mutant) in complex with laminarihexaose [54] shows, similar to the previous example, that two β-glucan chains bind within the single groove of the catalytic site, with four glucose units in one chain and five glucose units in the other chain. The two oligosaccharide chains are slightly twisted together and hence could essentially be part of a triple-helical β-1,3-glucan.

Anti-β-glucan antibody may also recognize structured β(1,3)-glucans. Antibody JoJ48C11 was generated against schizophyllan by an antibody phage display system. A crystal structure of this Fab fragment in complex with unbranched laminarihexaose yielded a partial electron density of the ligand [55]. If schizophyllan is assumed to have a triple helix, the coordinates fit well in the observed density. A β(1,6)-linked glucose residue is very important for antibody binding and the model of the complex does indeed suggest the involvement of a β(1,6)-linked glucose residue in the interaction. 

## 8. Effect of β(1,6)-Branching on β-Glucan Conformation

The effect of β1,6 branching on β-glucan conformation is of interest, because β(1,6)-branching significantly affects the biological activity of β-glucan [25]. Curdran has no side chain branching and assumes an insoluble triple helix in water. Schizophyllan, in contrast, is soluble in water. Hence, modification with a β(1,6)-linked glucose residue on every third β(1,3)-linked glucose residue increases the solubility. Hydrogen bonding and surrounding water molecules may play their roles in defining such behavior. The molecular dynamics study of Okobira et al., of triple helical β-glucan with and without β(1,6)-branching, shows that β(1,6)-branching affects several conformational properties [56]. As the population of side chains increases, the helical pitch decreases. The average pitch of curdran is 20.6 Å, while that of schizophyllan is 18.8 Å. Two types of hydrogen bonding exist for branched β-glucan: side chain–main chain and side chain–side chain. A small cavity with a diameter of 3.5 Å occurs within the triple helix of schizophyllan but not in curdran. Furthermore, β(1,6)-branching causes a tilt of the main-chain glucose residue with respect to the helix.

There is a paucity of experimental results that shed light on the conformation of branched β-glucans at the atomic level. One example is found in a crystallographic analysis of GH16 1,3(4)-β-glucanase (*Phanerochaete chrysosporium* laminarinase 16A), complexed with a β-glucan product of laminarin hydrolysis, i.e., Glcβ(1,6)-Glcβ(1,3)-Glcβ(1,3)-Glc [57]. In the crystal structure, two ligands are found: In the acceptor site is the tetrasaccharide, including a β(1,6)-linked side chain, while in the donor site are three β(1,3)-linked glucose residues (Figure 6). The β(1,6)-glucose residue in the acceptor site lies deep within the cleft and interacts with a main-chain glucose residue (+2 position) via a hydrogen bond. Experimental observations such as these, coupled with theoretical studies, will help to gain understanding of structure–function relationships of branched β-glucans, and the role of branching.

## 9. Summary and Future Perspectives

The triple helix structure of β-glucan was initially proposed from X-ray fiber diffraction studies and the atomic structure is now well-supported by several X-ray crystallographic analyses of short β(1,3)-glucan chains in complex with β-glucan chain binding proteins. These examples point to the importance of higher-order structures of β-glucan in the associated biological phenomena. However, several reports suggest that triple helical or higher-ordered structures are not essential or advantageous for the expression of certain biological activities [40,58,59]. Detailed information of how β-glucan is recognized and how downstream signaling occurs is still lacking. One difficulty in the study of structure–function relationships of β-glucans is because β-glucan from natural sources is chemically heterogeneous, e.g., in terms of chain length and branching. Also, determination of 3D structures of β-glucan remains challenging. Likely, advances will be greatest using chemically and conformationally defined β-glucan chains. These would be easier to obtain and analyze and should provide clearer structure–function relationships of β-glucan and the binding proteins. These are also expected to provide important clues to the development of β-glucan assay technology, covering a variety of structural characteristics, and that of new therapeutic drug treatment and monitoring of invasive fungal infections which can lead to sepsis.

## Figures and Tables

**Figure 1 ijms-22-01578-f001:**
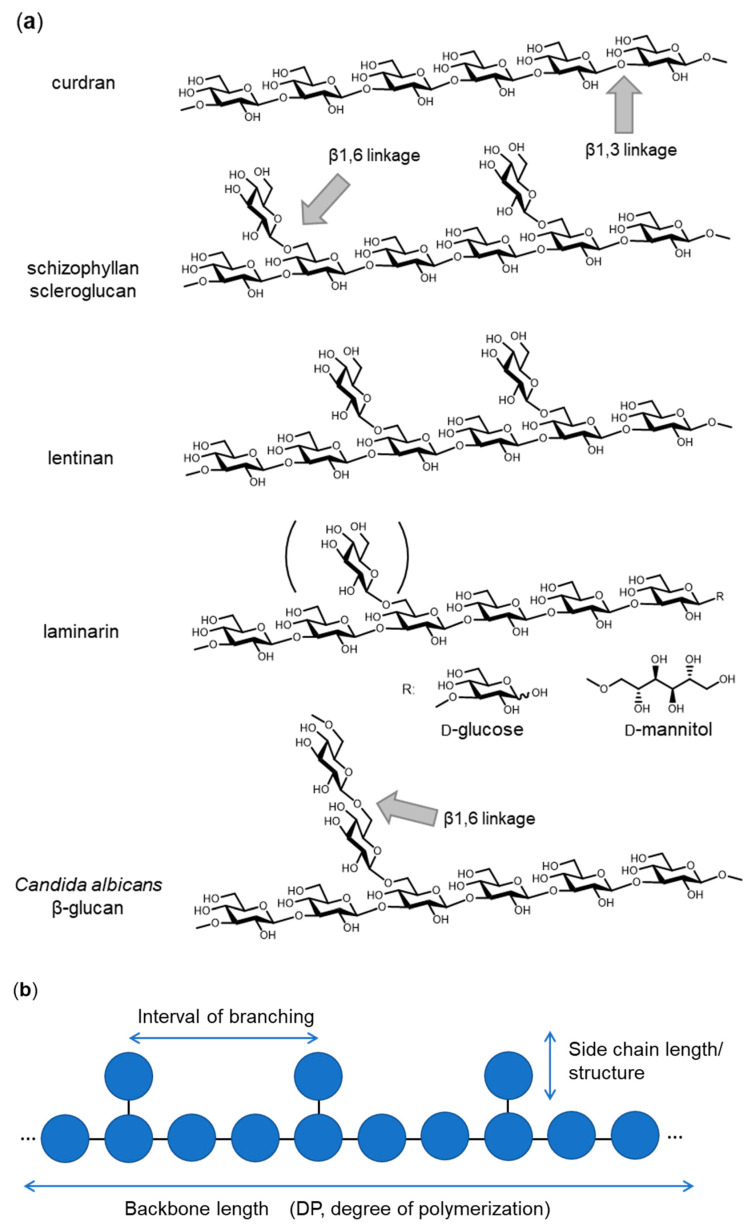
Chemical structure of representative β(1,3)-glucans. (**a**) Curdran from *Agrobacterium* species, schizophyllan from *Schizophyllum commune*, scleroglucan from *Sclerotium* species, lentinan from *Lentinula edodes*, laminarin from *Laminaria digitata* and *Candida albicans* β-glucan. (**b**) A schematic drawing of β-glucan with the key parameters (backbone length, interval of branching and side chain length/structure) that define the primary structure.

**Figure 2 ijms-22-01578-f002:**
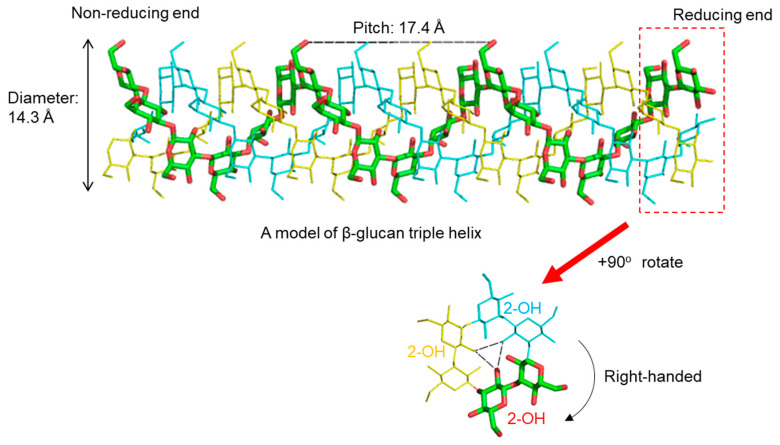
A model of right-handed triple helical β(1,3)-glucan based on X-ray diffraction data [11]. A pitch of helix is estimated as 17.4 Å and the diameter as 14.3 Å. Inter-strand hydrogen bonds are formed between 2-OH of glucose residues, which are perpendicular to the helix axis.

**Figure 3 ijms-22-01578-f003:**
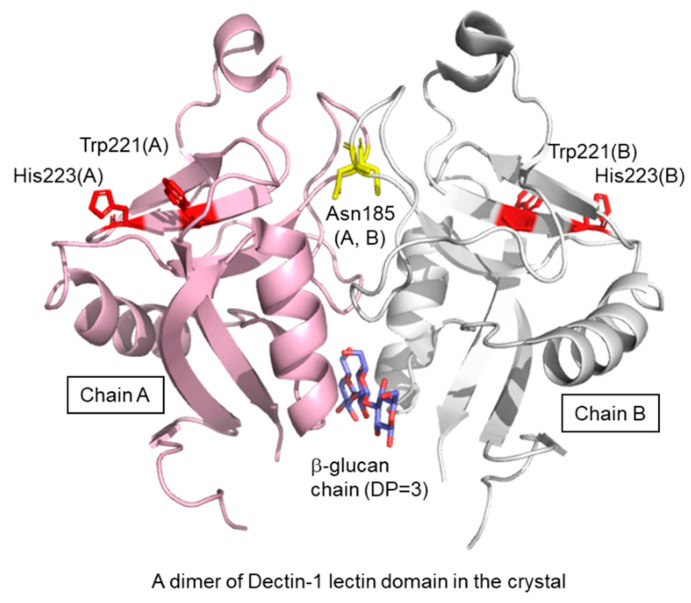
Crystal structure of dimeric Dectin-1 lectin domain trapping laminaritriose. Proteins are shown in ribbon model and chain A and chain B are colored in pink and white, respectively. Trapped laminaritriose, Trp221, His223 and Asn185 are in stick representation.

**Figure 4 ijms-22-01578-f004:**
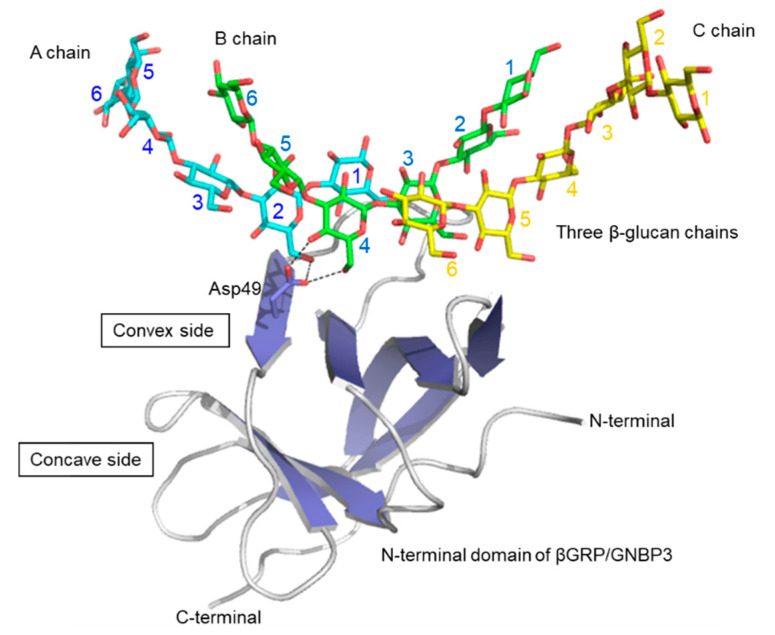
Crystal structure of *Bombyx mori* βGRP/GNBP3 N-terminal domain bound to three β-glucan chains [38]. Proteins are shown in the surface model and three laminarihexaose chains are in stick representation. As an example of β-glucan-interacting residues, Asp49 side chain is shown in stick representation, interacting with A and B chains.

**Figure 5 ijms-22-01578-f005:**
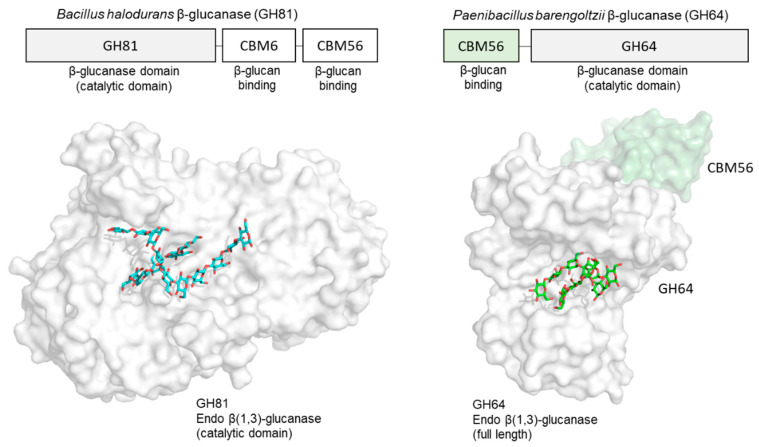
Crystal structure of endoglucanases bound to β-glucan chains. GH81 endo β-1,3-glucanase in complex with three β-glucan chains (DP = 10, 3, 2) derived from laminarin (PDB ID: 5T4G) (left) and GH64 endo β-1,3-glucanase in complex with two laminarihexaose chains (PDB ID: 5H9Y) (right). Proteins are shown in surface model and β-glucan chains in stick representation.

**Figure 6 ijms-22-01578-f006:**
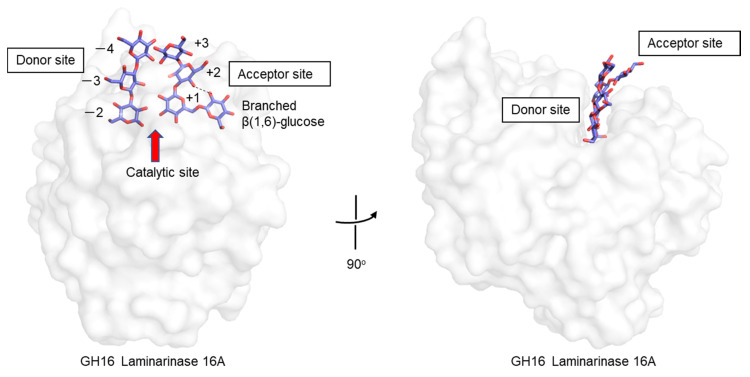
Crystal structure of a GH16 laminarinase in complex with products of laminarin. Proteins are shown in the surface model and β-glucan chains in stick representation. A hydrogen bond between β(1,6)-side chain and main chain glucose residue (+2 position) is shown as a dashed line.

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
