# Peer review of "3D Structural Insights into β-Glucans and Their Binding Proteins"

_ijms, 2021, doi:10.3390/ijms22041578_

Round 1
Reviewer 1 Report
This review is focusing on the interaction of b-glucans with b-glucan binding proteins, especially from the view point of 3D-strcutural characteristics. It is well written.
I felt that one important b-glucan binding protein from horseshoe crab, factor G, was missing. Interaction of factor G and b-glucan have been extensively analyzed and clinically applied. I would suggest to add sections about interaction of b-glucan with factor G.
Author Response
I felt that one important b-glucan binding protein from horseshoe crab, factor G, was missing. Interaction of factor G and b-glucan have been extensively analyzed and clinically applied. I would suggest to add sections about interaction of b-glucan with factor G.
Thank you very much for your suggestion. We added one section (section 6) titled “Factor G-beta-glucan interaction” at line 349.
Reviewer 2 Report
Interesting and important contribution to the current knowledge of glucan and its receptors. The chemical part is OK, but some problems persist:
1) Paper constantly jumps from invertebrate to vertebrate receptors and back, making reading difficult
2) only Dectin-1 was described. CR3 receptor, as another highly important glucan receptor, has to be fully described.
3) Sentences such as "humans and other mammals do not have a system...." is unclear. Actually vertebrates do not synthesize glucan, but why is this unclear sentence the second sentence in a manuscript focused mostly on invertebrate receptor? In addition, it is not really relevant, as glucan is also recognized by plants.
4) chemical structure of 5 glucans is shown. It is important to either show many more different glucan or add that this is just for orientation only and explain why these particular glucans were used.
Author Response
1) Paper constantly jumps from invertebrate to vertebrate receptors and back, making reading difficult
Thank you. We have re-ordered the sections from vertebrate to invertebrate to make it easier to read.
2) only Dectin-1 was described. CR3 receptor, as another highly important glucan receptor, has to be fully described.
We added Section 4 on CR3 (at line 248) to describe the key information reported so far.
3) Sentences such as "humans and other mammals do not have a system...." is unclear. Actually vertebrates do not synthesize glucan, but why is this unclear sentence the second sentence in a manuscript focused mostly on invertebrate receptor? In addition, it is not really relevant, as glucan is also recognized by plants.
We appreciate the comments. To make it clear, the sentences are modified in the abstract section (line 9) and introduction part (line 22).
4) chemical structure of 5 glucans is shown. It is important to either show many more different glucan or add that this is just for orientation only and explain why these particular glucans were used.
We did not intend to cover all the beta-glucans, rather to show representative ones to orient the readers in the idea of length, interval, and branching of beta-glucans. We modified the legend of Figure 1 to “Chemical structure of representative beta(1,3)-glucans…. “, at line 93.
Round 2
Reviewer 2 Report
This is much improved version. All requests were incorporated into the text.